# Sarcopenia: Etiology, Nutritional Approaches, and miRNAs

**DOI:** 10.3390/ijms22189724

**Published:** 2021-09-08

**Authors:** Roberto Cannataro, Leandro Carbone, Jorge L. Petro, Erika Cione, Salvador Vargas, Heidy Angulo, Diego A. Forero, Adrián Odriozola-Martínez, Richard B. Kreider, Diego A. Bonilla

**Affiliations:** 1Department of Pharmacy, Health and Nutritional Sciences, University of Calabria, 87036 Rende, Italy; erika.cione@unical.it; 2Galascreen Laboratories, University of Calabria, 87036 Rende, Italy; 3Research Division, Dynamical Business & Science Society, DBSS International SAS, Bogotá 110311, Colombia; jlpetro@dbss.pro (J.L.P.); salvadorvargasmolina@gmail.com (S.V.); dabonilla@dbss.pro (D.A.B.); 4Faculty of Medicine, University of Salvador, Buenos Aires 1020, Argentina; 5Research Group in Physical Activity, Sports and Health Sciences (GICAFS), Universidad de Córdoba, Montería 230002, Colombia; 6Faculty of Sport Sciences, EADE-University of Wales Trinity Saint David, 29018 Málaga, Spain; 7Grupo de Investigación Programa de Medicina (GINUMED), Corporación Universitaria Rafael Núñez, Cartagena 130001, Colombia; heidy.angulo@curnvirtual.edu.co; 8Health and Sport Sciences Research Group, School of Health and Sport Sciences, Fundación Universitaria del Área Andina, Bogotá 111221, Colombia; dforero41@areandina.edu.co; 9Sport Genomics Research Group, Department of Genetics, Physical Anthropology and Animal Physiology, University of the Basque Country UPV/EHU, 48940 Leioa, Spain; adrianodriozola@gmail.com; 10kDNA Genomics^®^, Joxe Mari Korta Research Center, University of the Basque Country UPV/EHU, 20018 Donostia-San Sebastián, Spain; 11Exercise & Sport Nutrition Lab, Human Clinical Research Facility, Texas A&M University, College Station, TX 77843, USA; rbkreider@tamu.edu; 12Research Group in Biochemistry and Molecular Biology, Universidad Distrital Francisco José de Caldas, Bogotá 110311, Colombia

**Keywords:** mitochondria, aging, protein, muscle, nutritional supplements, MyomiR

## Abstract

Sarcopenia, an age-related decline in skeletal muscle mass and function, dramatically affects the quality of life. Although there is a consensus that sarcopenia is a multifactorial syndrome, the etiology and underlying mechanisms are not yet delineated. Moreover, research about nutritional interventions to prevent the development of sarcopenia is mainly focused on the amount and quality of protein intake. The impact of several nutrition strategies that consider timing of food intake, anti-inflammatory nutrients, metabolic control, and the role of mitochondrial function on the progression of sarcopenia is not fully understood. This narrative review summarizes the metabolic background of this phenomenon and proposes an integral nutritional approach (including dietary supplements such as creatine monohydrate) to target potential molecular pathways that may affect reduce or ameliorate the adverse effects of sarcopenia. Lastly, miRNAs, in particular those produced by skeletal muscle (MyomiR), might represent a valid tool to evaluate sarcopenia progression as a potential rapid and early biomarker for diagnosis and characterization.

## 1. Introduction

According to the World Health Organization (WHO), life expectancy has increased by 5 years since 2000 in developed countries [1]. This increase in life expectancy leads to aging and, in turn, to physical and cognitive decline. Current research shows that physical exercise can be a decisive protective factor for both functional decline and negative body composition changes during aging [2,3]. Resistance and endurance exercise have been shown to contribute to the improvement of cognition [4], as well as related psychological and social factors [5]. However, nutrition and eating strategies also play an essential role in preventing and treating functional limitations in the elderly population. For example, muscle mass declines by approximately 3–8% for each decade after 30 years, and this percentage increases significantly in people older than 60 years [6].

Sarcopenia is a condition characterized by a progressive and generalized loss of skeletal muscle mass and function with an increased risk of adverse outcomes such as disability, metabolic dysfunction, poor quality of life, and death [7]. Even though sarcopenia is mainly associated with the aging process seen in the elderly, there are several other populations at risk due to lifestyle decisions or pathological states. These include sedentary, immobilization, malnutrition, diabetes, obesity, and other acute or chronic inflammatory diseases that could promote the loss of muscle mass [8]. Moreover, the loss of muscle mass could also negatively affect the outcomes of those conditions [9,10]. In the last decade, three different consensus papers were published, giving both a definition of sarcopenia and diagnostic criteria. According to the European Working Group on Sarcopenia in Older People (EWGSOP), the presence of low skeletal muscle mass (e.g., DXA, Anthropometry) and either low muscle strength (e.g., handgrip, isokinetic) or low muscle performance (e.g., walking speed, muscle power), which is often related to the most advanced stages of sarcopenia, are the criteria for the diagnosis of sarcopenia [9]. For both the European Society for Clinical Nutrition and Metabolism Special Interest Groups (ESPEN-SIG) and the International Working Group on Sarcopenia (IWGS), only the loss of muscle concomitant with a loss of muscle strength (which could also be assessed by walking speed) represents the recommended parameters. The IWGS also highlights that the loss of muscle mass could be alone or in conjunction with an increase in fat mass [10,11].

Thus, the diagnosis of sarcopenia can then be carried out by assessing the following parameters in the elderly (>65 years): (i) if walking speed is below 0.8 m/s at the 4 m walking test, and there is a low amount of muscle mass (i.e., a percentage of muscle mass divided by height squared is below two standard deviations of the normal young mean (<7.23 kg/m^2^ in men and <5.67 kg/m^2^ in women) as defined using dual-energy X-ray absorptiometry [12], or (ii) if the walking speed at the 4 m walking test is higher than 0.8 m/s, the hand-grip strength should be tested. If this last value is lower than 20 kg in women and 30 kg in men, the muscle mass must be analyzed [12].

While sarcopenia is mainly observed in the elderly, it can also develop in young adults [10]. A clear cause of sarcopenia cannot always be identified. Thus, the category of primary and secondary sarcopenia may be useful for choosing the best strategy to delay the progression. Primary sarcopenia is considered when there is no other evident cause rather than the aging process itself, while sarcopenia can be considered secondary when one or more of the following causes are evident [10]: (i) activity-related causes (bed rest, sedentary lifestyle, reconditioning, or even zero-gravity conditions seen in astronauts); (ii) nutrition-related causes, as a result of an inadequate dietary intake either energy or protein, malabsorption of nutrients or gastrointestinal disorders, and anorexigenic medication; (iii) disease-related causes, where there is a cross-talk between muscle mass and other organs that could lead to sarcopenia, such as inflammation, endocrine diseases, advanced organ failure, and malignancy. In this sense, sarcopenia is linked to the prognostic state of some pathologies, e.g., in those that affect the intestinal function and, as aforementioned, the absorption of nutrients as reported in abdominal hernias [13] and inflammatory bowel disease (IBD) [14]. Furthermore, sarcopenia has been recently related to the prognosis of various tumors [15,16,17,18], which leads to a monitoring status at the initial stage, as well as during and after chemotherapy. This narrative review aims to summarize the key molecular mechanisms of sarcopenia, the recent and future directions of the microRNAs for diagnosis and prognosis, and several strategies with an emphasis on nutrition to counteract the progression of this condition.

## 2. Mechanisms

The mechanisms of sarcopenia are not clearly defined. Well-described risk factors for sarcopenia include age, sex, physical activity level (e.g., resistance exercise is particularly effective for slowing the age-related loss of musculoskeletal tissue), and nutrition habits. For example, a deficient energy and/or protein intake will contribute to a reduction in both the amount and the functionality of the skeletal muscle. Furthermore, sarcopenia is associated with major comorbidities such as obesity, osteoporosis, type 2 diabetes, and insulin resistance [9]. These comorbidities (acute and/or chronic) may lead to reduced physical activity characterized by periods of bed rest, and the generation of a proinflammatory cytokines plays an important role in activating the muscle proteolysis machinery. Individuals with an active lifestyle have more lean body mass and muscle mass when aged [7,18].

Hence, sarcopenia is a universal phenomenon with a complex and multifactorial etiology. At the molecular level, sarcopenia results from a disproportionate increase in muscle protein breakdown and/or decrease in muscle protein synthesis [11]. Moreover, data clearly indicate that the loss of muscle fibers is accompanied by a reduction in the number of motor neurons, which is the main factor that contributes to the decline in muscle strength. However, whether motor neuron loss is a cause or a consequence of muscle fiber deficits has not been definitively established in humans [19]. Neuromuscular junctions in muscle fibers show a variety of alterations, including axonal swellings, sprouting, synaptic detachment, withdrawal of axons from postsynaptic sites, and fragmentation of the acetylcholine receptors [20]. Such neuronal changes have been proposed to play a major role in the age-related loss of muscle mass and function.

### 2.1. Muscle Protein Synthesis

The synthesis and degradation of proteins are two intimately coupled mechanisms. Most of the molecular pathways involved concomitantly regulate both processes; thus, when protein synthesis is induced, the degradation is suppressed and vice versa. Notwithstanding, this control appears to be a compensatory mechanism to limit energy expenditure for the production of novel proteins under catabolic conditions [20]. Anabolic hormones, certain nutritional compounds, and muscular activity are positive effectors of the receptor tyrosine kinases and the phosphatidyl inositol 3-kinase (PI3K)/Akt pathway [21]. This metabolic pathway stimulates muscle protein synthesis through the activation of the mechanistic target of rapamycin complex 1 (mTORC1) but also inhibits catabolic pathways via phosphorylation of the forkhead box transcription factors (FOXO). mTORC1 is known as a key regulator in controlling skeletal muscle mass through stimulation of protein synthesis, which is required for cell growth, proliferation, and differentiation [22], but it is also implicated in the regulation of general autophagy and mitophagy [23]. Phosphorylated FOXO is unable to enter into the cell nucleus, which reduces the expression of the E3 ligase, Atrogin I and RING finger-1 (MuRF1), which subsequently prevents protein degradation via the ubiquitin-proteasome system (UPS) and autophagic-lysosomal pathways [11,22]. The UPS is mainly responsible for the degradation of short-living proteins, whereas autophagy regulates the breakdown of long-living proteins and organelles. However, a large body of evidence suggests that upregulating both pathways also plays an important role in skeletal muscle wasting [24].

The aging process results in a significant decline in different anabolic hormones. In females, estrogen and other sex hormones decline after menopause [9,25]. Furthermore, plasma insulin-like growth factor 1 (IGF-1) concentrations are significantly associated with age in both men and women. Circulating IGF-1 plays an active role in processes of protein synthesis via activation of the mTORC1 pathway, as well as in the regulation of growth hormone (GH) secretion through a negative feedback mechanism [26,27]. Both testosterone and GH are powerful anabolic agents that promote muscle protein synthesis and subsequent muscle mass accretion [28]. Moreover, estrogen may play a significant role in stimulating muscle repair and regenerative processes, including the activation and proliferation of satellite cells [18,19,20,21,22,23,24,25,26,27,28,29].

### 2.2. Oxidative Stress

Given the complex etiology of sarcopenia and the different factors affecting its development, addressing only energy balance and protein intake and quality could be a very simplistic approach [11]. Sarcopenia is characterized by an inflammatory status, mainly in muscle tissue, which shares molecular insults with other chronic diseases that have a strong disturbance in mitochondrial function and central/peripheral circadian rhythms [30,31]. It has been shown that aging increases oxidative stress, which could negatively affect mitochondria quality control [32]. Mitochondrial content, structure, network, and function are crucial factors for maintaining both skeletal muscle mass and functionality. In fact, when mitochondria quality is impaired, the hormetic generation of reactive oxygen species (ROS) is disrupted, leading to a decline in cellular function and general health [33]. Aging also increases the ROS production and/or decreases antioxidant enzymatic levels in muscle and brain. Thus, the mitochondrial degeneration is associated with impaired energy generation and ROS production, which might be the primary initiators for the resulting alteration of the phenotype in sarcopenia [34,35]. This disbalance in ROS production correlates with the increase in inflammatory mediators, such as tumor necrosis factor α (TNFα), interleukin 6 (IL-6), nuclear factor kappa B (NF-κB) and C-reactive protein (CRP). These molecules are associated with aging and can activate several transcription factors that modulate gene expression and regulate muscle wasting via the UPS [31,36].

Even though the exact reaction mechanism is still under debate, several theories have been proposed. Firstly, the excess of ROS generation induces damage of the mitochondrial membrane, DNA, and proteins. Secondly, the opening of the mitochondria permeability transition pore (mPTP) induces a further loss of mitochondrial membrane potential and releases mitochondrial content into the cytosol to initiate apoptosis in a tissue-dependent manner [34]. Consequently, there is an accumulation of damaged mitochondrial with an impaired energy and ROS generation, which results in the apoptotic response [34]. Lastly, ROS generation induces overexpression of inflammatory markers and, hence, changes in transcription factors and gene expression that affect proteostasis [31]. Thus, both exercise and nutritional approaches may attenuate sarcopenia by targeting mitochondrial health and function [34,37,38,39].

## 3. The Importance of MicroRNAs

Low-grade inflammation is defined as an inflammatory state that cannot be determined by classical standards such as CRP or where an increase in IL-6 and TNFα is sometimes noted. It is established in sarcopenia [40], as well as in various pathological states such as obesity [41], cancer [42], polycystic ovary syndrome [43], and osteoarthritis [44]. In addition to the afflicted tissue, the latent and chronic inflammation state makes the whole organism less efficient by triggering chain reactions through crosstalk between tissues. For instance, Wang et al. [45] proposed that, during insulin resistance (associated with obesity), there is a phenomenon of latent inflammation that aggravates the extent and progression of sarcopenia. It is reasonable to think that an inflammatory state is also present in sarcopenia; therefore, if it is a secondary state or there are chronic comorbidities, there will be a further contribution to the inflammatory state. Hence, mitigating inflammation could be beneficial in the management of sarcopenia. Kwang Byun et al. [46] correlated systemic inflammation present in chronic obstructive pulmonary disease (COPD) with sarcopenia. Similarly, Dalle et al. [47] demonstrated that three apparently different pathological states (i.e., type II diabetes, osteoarthritis, and COPD) are associated with an inflammatory state and lead to sarcopenia. Interestingly, a latent inflammatory state such as IBD also correlates to a greater onset of sarcopenia [14,48]. This status of low-grade inflammation has been a constant focus of research in order to identify new potential biomarkers.

MicroRNAs (miRNAs) are a unique class of short endogenous nucleotides sequences (around 15–30 bases). They are single-stranded noncoding RNAs capable of modulating gene expression by binding to the complementary regions of the 3′UTR sequence of specific mRNA targets, resulting in the inhibition of protein synthesis (translation) and/or mRNA degradation. This peculiar regulatory capability makes them crucial for normal development in all living beings [49,50]. miRNAs are present in all tissues and body fluids [51,52]. One important characteristic of the skeletal muscle is a group of miRNAs, identified as myomiRs [53,54,55], which seem to have a central role in the regulation of skeletal muscle plasticity by coordinating changes in fiber type and muscle mass in response to different contractile activity. Like every tissue, skeletal muscle also expresses miRNAs. In particular, the pool of these molecules is defined as myomiRs and is related to the differentiation of satellite cells, the maintenance of physiological trophism, the switch between fibers, and the development and conservation of muscle mass in response to physical exercise [53]. For example, Soares and colleagues [54] demonstrated an important regulatory action of a group of miRNAs on the progression of muscle atrophy. Brown et al. [56] showed that a pool of miRNAs (miR-23a, miR-182, miR-486, miR-206, miR-21, miR-27, and miR-128) are strong regulators of muscle size via the FOXO1 pathway, PTEN genes and translational regulation, and myostatin signaling. MyomiRs are secreted via exosomal vesicles, circulate in the bloodstream, and serve as regulators/communicators in proximal muscle tissue and even fat cells [54,55].

As aforementioned, sarcopenia is certainly the result of several factors, and its etiopathogenesis is still not well identified. For this reason, the identification of miRNAs might contribute to better understand this phenomenon, although the description of the myomiR profile is in its infancy. This group of miRNAs is potentially involved in the regulation of the satellite cell differentiation, the general proteostasis, the structure and type of muscle fibers, mitochondria and oxidative stress metabolism, the neurodegeneration process, and the infiltration of adipocytes into skeletal muscle tissue [55,57].

### 3.1. Satellite Cell Regulation

The differentiation of satellite cells is a fundamental process for the maintenance of muscle trophism. In this sense, certain miRNAs (miR-1, miR-206, and miR-486) have been identified to regulate cell survival and proliferation [58,59]. miRNAs have been sown to downregulate the MyoD and paired-box transcription factor (e.g., Pax3) pathways, which result in an inhibition of apoptosis, thereby increasing or maintaining muscle mass. These results have been seen in preclinical research; hence, it is highly plausible they have similar action mechanisms in humans (considering well-conserved metabolic signatures) [60]. It is noteworthy that both endurance and resistance training impact myomiRs, particularly those involved in skeletal muscle allostasis. For example, certain miRNAs regulate the expression of growth factors (miR-29, miR-422-5p, and miR-143-3p), cell-cycle regulation (Let-7b and Let-7e), and myocyte differentiation (miR-139, miR-155, miR-501-3p, and miR-29) [61].

### 3.2. Proteostasis

At the moment, there are not many studies in humans. For example, Connors et al. [62] reported an increase in miR-424-5p during a decrease in protein synthesis at the skeletal muscle level with a consequent loss of muscle mass. It is well known that the control of protein metabolism is mediated by several miRNAs: miR-199, miR-125b, and miR-195 regulate hormones such as insulin and IGF-1; miR-432, miR-675-3p, miR-26a, miR-29, and miR-199-3p regulate signal transduction within the muscle cell; miR-27 and miR-128 serve as myostatin regulators; miR-129c, miR-23c, miR-27a, and miR-35 regulate protein catabolism [57].

### 3.3. Size and Type of Muscle Fiber

Some miRNAs seem to play a decisive role in the structure of muscle fibers. In particular, miR-23a and miR-182 regulate what are called iatrogenic genes (i.e., those who oversee the muscle atrophy program), as they appear to be able to restore drug-induced atrophy. Similarly, miR-21 and miR-206 are capable of acting on the regulation of atrophy. On the other hand, miR-27a seems to downregulate myostatin, thus favoring muscle turnover in a positive balance [55,63,64].

### 3.4. ROS and Mitochondria

As mentioned previously, ROS are produced mainly from mitochondria as a consequence of inflammatory stimuli to mediate apoptosis. Both phenomena are present in the development of sarcopenia. A group of approximately 400 miRNAs called mitomiRs are particularly linked to correct mitochondrial function and structure [65]. Mir-340-5p and miR-206 showed a regulatory action on nuclear respiratory factor-1 (Nrf2) which is one of the main modulators of ROS synthesis; therefore, monitoring these miRNAs could be a potential strategy to evaluate ROS production [57]. According to the hypothesis of Rippo et al. [66], some miRNAs (miR-181a, miR-34a, and miR-146a) might belong to a so-called ‘*inflammaging*’ phenotype (i.e., age-related inflammation and/or recurrent inflammatory stimuli such as chronic and/or subclinical due to obesity).

### 3.5. Fat Infiltration

The accumulation of ectopic fat within musculoskeletal tissue is one of the consequences of sarcopenia, which contributes to the low-grade inflammatory state and, hence, to imbalance and metabolic disturbances [57,67]. Although found in rats and still not confirmed in humans, miRNAs have a key role in regulating this process of fat infiltration; miR-133 and miR-499 regulate the differentiation process of brown adipocytes; miR-23 limits the infiltration of adipocytes into muscle tissue by inhibiting the action of platelet-derived growth factor receptor α (PDGFRα); miR-130b regulates mitochondrial allostasis and energy expenditure by acting on proliferator-activated receptor gamma coactivator 1 alpha (PGC-1α) [68,69,70].

### 3.6. miRNAs as a Therapy Tool and Biomarker

We have highlighted how miRNAs are present in all biological fluids as microvesicle-mediated secreted molecules, are partially stable, and are, therefore, potential candidates for use as biomarkers (Figure 1). Liu et al. [71] proposed miR-146a and miR-486 as possible biomarkers of sarcopenia in the elderly population. miR-486 is related to the response to physical exercise, but further research is warranted to validate its use. Future studies could verify if other miRNAs linked to obesity-related sarcopenia can be used to monitor the age-related muscle loss. Furthermore, more research is needed to establish if myomiRs can be screened in biological fluids other than blood (e.g., saliva or urine). Lastly, an intriguing hypothesis is that regulating the action of miRNAs with antagomiRs, nucleotide sequences capable of pairing with miRNAs and preventing their action, might serve as a potential therapeutic tool; this is being evaluated in phase 1 and 2 clinical trials for other pathologies (nephritis and fibrosis) [57].

## 4. Counteracting Strategies

### 4.1. Physical Exercise with Emphasis on Resistance Training

The fundamental role of physical exercise in countering the progression of sarcopenia, associated or not with obesity, is now evident. Clinical research has confirmed the effectiveness of physical exercise, both cardiovascular and resistance training [72]. As we described before, myomiRs are closely related to an optimal condition of muscle tissue and show an important role as signaling molecules that mediate physiological adaptations to exercise training. Furthermore, these miRNAs change differently in response to cardiovascular, resistance, or combined exercise (e.g., miR-1, miR-133a/b, miR-206, miR-499a-5p, and miR-486), but with no apparent difference in response between young and old men [73]. There is strong evidence that strength training is one of, if not the most, effective interventional strategy to enhance muscle mass and strength in the elderly; thus, it can be used for treating, slowing, and/or preventing sarcopenia and dynapenia [74]. Resistance training enhances physiological adaptations of the neuromuscular system, which positively affects the muscle strength. Maximal motor unit discharge rates increased 49% in older adults that followed only 6 weeks of a high-intensity progressive strength training program [75]. Moreover, muscular factors independent of muscle mass, such as fascicle length and tendon stiffness, have also been observed to improve (10% and 64%, respectively) following resistance training in older adults [76]. Moreover, resistance training is also a powerful stimulus for muscle protein synthesis, which leads to an increase in muscle mass. In this sense, an increase in the cross-sectional area of the thigh muscle (+4.6%) has been reported in mobility-limited older adults after 24 weeks of a resistance training program in conjunction with protein supplementation [77]. Thus, there is a strong consensus in this regard. A review of 121 trials including over 6700 participants concluded that ‘*progressive resistance training is an effective intervention for improving physical functioning in older people, including improving strength and the performance of some simple and complex activities*’ [78]. The authors reported a large positive effect on muscle mass, strength, and functionality. Additionally, high-intensity resistance training is associated with greater benefits in muscle strength with an average improvement of 5.3% after each incremental in exercise intensity from low intensity (<60% 1-RM), to low/moderate intensity (60–69% 1-RM), moderate/high intensity (70–79% 1-RM), and high intensity (≥80% 1-RM) [75]. High-intensity resistance training has been reported to be well tolerated in older adults, particularly when a proper progression is applied [79,80], although intensities between 65% and 75% 1-RM can be sufficient to promote significant adaptations [81]. Higher resistance training volumes are associated with greater improvements in lean body mass after controlling for a variety of confounders (e.g., age, study duration, sex, and training intensity and frequency) [82]. With regard to strength training frequency, 2–4 days per week are commonly recommended with training typically being performed on alternating days (e.g., Monday, Wednesday, and Friday) [81]. A well-prescribed resistance training program should also include exercises targeting all major muscle groups, but emphasis on lower limbs is recommended. Significant improvements in muscle strength and size have been reported in training programs that include 1–3 sets per exercise [83] with an adjustment of the numbers of repetitions that considers the maximum number that can be performed with a given intensity (max effort) [84]. Even in delicate conditions such as osteoarthritis or spondylarthritis, strength training can give excellent results [85,86]. Lastly, in order to continually reach improvements in mass, strength, and functional capacity, it is key to consistently incorporate progression and variation into the program. Every training variable can be adjusted over time considering the training experience of the subject and the adaptation rate on a case-by-case basis [81,87]. It is worth mentioning that it is not always possible to practice physical exercise, particularly strength training, for bedridden subjects and/or long-term patients due to chronic diseases such as cirrhosis, COPD, or severe renal insufficiency up to dialysis.

### 4.2. Nutrition and Supplementation

Maintenance of energy balance is crucial during a period of muscle disuse, but simply overfeeding does not further attenuate muscle atrophy since this merely increases adipose tissue. The key factor behind an accelerated loss of muscle tissue during a period of reduced food intake may not be the lower energy intake per se but more specifically the reduction in protein intake [88]. Barbera et al. [89] suggested some types of nutrients that would be able to influence the expression of myomiRs, regardless of the physical activity. Importantly, the intake of essential amino acids could positively impact miR-1, miR23a, miR208b, miR-499, and miR-27a, which have a positive effect on myocyte regeneration, proliferation, and differentiation [90]. In addition, resveratrol could regulate the differentiation of muscle cells and the activation of the PGC-1α through the positive modulation of miR-21 and miR27b and the downregulation of miR-133b, miR30b, and miR-149. Other nutrients have also been found to modulate these molecules such as albumin, palmitic acid, vitamin D, and fructose [91]. PGC-1α is a critical cofactor for mitochondrial biogenesis that it is mainly activated by the AMP-activated protein kinase (AMPK) pathway. AMPK is one of the main energy sensors (perturbations in ATP/ADP ratio) that regulate energy metabolism (e.g., protein synthesis, as an energy cost process) [92]. High levels of PGC-1α are associated with muscle mass sparing during sarcopenia, possibly by means of a reduction in the protein breakdown via FOXO inhibition (with no changes in protein synthesis) [21]. It has been shown that FOXO induces the expression of atrogin-1 and MuRF1 under conditions of energy stress in myofibers, but activation of PGC-1α could attenuate the negative regulation of proteostasis [21,93].

Starvation and aggressive hypocaloric diets have been reported as deleterious to the muscle mass and function, especially when the protein needs are not achieved [94,95]. This might be due to inhibition of the mTORC1 pathway, as demonstrated after some weeks of low-carbohydrate high-fat (LCHF) diets [96]. Even though extreme nutrient and energy deprivation induces autophagy, a mild carbohydrate restriction may result in a favorable impact on sarcopenia outcomes [21,97,98]. In fact, caloric restriction might confer lifespan and health benefits and, therefore, it is not surprising that intermittent and periodized caloric restrictions (e.g., alternate-day fasting or intermittent fasting) might be suitable as a counteracting strategy for sarcopenia [99,100]. Thus, certain biological elements might prevent the excessive activation of UPS via negative regulation of pro-atrophy transcription factors without modifying the translational process. From a nutritional standpoint, adequate protein intake and certain antioxidants (e.g., secondary metabolites) could modulate the muscle protein synthesis and breakdown. The subsections below summarize relevant findings in this regard.

#### 4.2.1. High-Protein Diet

Older people have a diminished myofibrillar protein synthesis response to protein intake, which may have a strong influence on the progression of sarcopenia, and it is exacerbated in elderly population with obesity. This age-related muscle ‘anabolic resistance’ is more evident in response to low or moderate protein intake which is common in the diet of older individuals [101]. In addition, the current recommended dietary protein intake of 0.8 g/kg/day might not be sufficient for preserving muscle mass and quality on a long-term basis [102,103,104]. In a recent review and meta-analysis, carried out on older subjects with overweight and obesity, it was concluded that protein intakes ≥1.0 g/kg/day have a greater protective effect on the loss of lean tissue than lower intakes. It is worth mentioning that subjects were 50 years old, considering that, as age advances, the amount of protein intake may become more essential [105,106]. Moreover, feeding is a critical modulator of the inner biological clock; therefore, both the timing and the type of food can be important [107,108]. Erratic eating patterns can disrupt the temporal coordination of metabolism and physiology, which is associated with chronic diseases that are also characteristic of aging such as sarcopenia [109,110]. Therefore, timing is crucial to regulate autophagy and mitophagy in more metabolically sensitive populations such as older adults.

#### 4.2.2. Nutritional Supplements

Several nutrients have shown potential modulatory effects on muscle protein synthesis and general physical health in elderly population [111,112,113] Table 1 briefly describes the most relevant in practice.

#### 4.2.3. Antioxidants and Inflammation

A reduction in ROS production and an increase in molecular ROS scavengers are two related processes. In mammalian cells, the master regulator of oxidant defense is the Nrf2 pathway [152]. The Nrf2 transcription factor induces the expression of antioxidant enzymes that protect against oxidative damage through binding to antioxidant response elements [152,153]. This important cellular mediator together with NF-κB is strongly influenced by polyphenols from the diet, as evidenced by the variation of miRNAs in response to diet [154,155]. Moreover, there is some evidence that certain molecules could impact the enzymatic and nonenzymatic regulators that protect the cell from oxidative damage, including superoxide dismutase, glutathione peroxidase, and glutathione [156,157]: (i) whey protein has been shown to promote an increase in intracellular glutathione levels [158], in addition to serving as a practical strategy for increasing protein intake and attenuating low-grade inflammation [159]; (ii) *N*-acetylcysteine, an acetylated form of cysteine, can properly maintain the availability of cysteine in the blood, which is the rate-limiting substrate for glutathione synthesis [156,160]; (iii) sulforaphane, an isothiocyanate compound derived from the diet (mainly in cruciferous vegetables), can activate the Nrf2 pathway [157,161]; (iv) creatine may control ROS generation by acting as a zwitterion that stabilizes the phospholipid membrane bilayer [135,162], possibly via regulation of mitochondrial respiration and UCP1-independent thermogenesis (i.e., futile creatine cycling) [163]; (v) vitamin D has a marked ability to control the expression of Nrf2 and the antiaging protein Klotho [164]. Vitamin D also regulates the function of the mitochondrial respiratory chain and may be important in the modulation of ROS production and lipotoxicity [165,166,167].

On the other hand, LCHF and ketogenic diets have been shown to be effective in regulating inflammation, while also modulating the miRNAs involved in the regulation of anti-inflammatory and antioxidant mediators [154,168,169,170]. These diets usually contain less than 20% of energy from carbohydrate (or less than 30 g per day) with more than 50% of energy from fat and variable amounts of protein [171,172], which leads to ketosis after the liver oxidizes high concentrations of non-esterified fatty acids into ketone bodies (i.e., 3-hydroxybutyrate, acetoacetate, and acetone). These are energy sources and signaling molecules in several tissues including the heart, the brain, the liver, and skeletal muscle. It is worth noting that LCHF can stimulate the AMPK pathway, thereby, activating PGC-1α, which leads to mitochondrial biogenesis [172,173], modulation of oxidative stress by activating UCP function, and increasing antioxidant capacity [174]. However, caution must be taken in adopting a LCHF diet, given that preclinical evidence has demonstrated a downregulation of the IGF1/Akt/mTORC1 pathway [173], and more research is needed with regard to sarcopenia. The restriction of carbohydrates should not be chronic and must be accompanied by a sufficient amount of protein intake. Fat intake should include adequate amounts of omega-3 polyunsaturated fatty acids since these are useful to manage low-grade inflammation, while the ratio between omega-3 and omega-6 should also be taken into account [175,176,177]. It should be considered that this nutritional program may not be applicable in subjects with liver cirrhosis, and it should be carefully evaluated in relation to renal function; in cases of application with impaired or absent renal function (dialysis), the protein–lipid ratio is unbalanced toward the latter.

## 5. Practical Nutritional Recommendations

High-quality protein intake is essential, given that musculoskeletal tissue keeps its ability to respond to increased plasma essential amino acids. However, protein intake should be augmented in the elderly to preserve muscle loss and fulfil physiological requirements [101,102,104] (Table 2). Each meal should ensure ~25 g of protein (0.3–0.4 g protein per kilogram of body mass,) although a higher amount is recommended after resistance exercise (~0.5 g protein per kilogram of body mass) [178,179]. Suggested sources include poultry, fish and seafood, eggs, dairy products, meat, cereals, and legumes [104].

With regard to carbohydrate and fat intake, the most important feature about their nutrient distribution is diet adherence. Although a mild reduction in carbohydrates has been suggested [180], further research is warranted. Recommended sources include fruits, legumes, and whole cereals [181,182]. Furthermore, it has been suggested that consuming carbohydrates during the daytime has a better glycemic response [110], while a low-carbohydrate dinner with higher fat and protein intake might be utilized [111]. Fat intake should be adjusted to meet daily caloric needs with approximately 1–1.5 g per kilogram of body mass. Main sources encompass olive oil, nuts, seeds, fish, and seafood, aiming to reach at least 250 mg of combined EPA and DHA per day [177]. Lastly, the post-exercise intake of high-quality protein should be of 0.5 g per kilogram of body mass (either food or whey protein supplement), and it can be combined with 0.1 g of creatine monohydrate per kilogram of body mass [135]. Particular attention should be paid to the contribution of polyphenols in the diet, as it has been seen that they have a positive action on the regulation of NRF2 (also highlighted by the action on some miRNAs), in addition to acting directly as antioxidants and anti-inflammatories; therefore, the regular intake of fruit and vegetables and/or their integration are recommended [183,184].

## 6. Conclusions

From a nutritional standpoint, both the amount/quality of protein and the use of some nutritional supplements (e.g., HMB and creatine monohydrate) can be considered as the most important factors to counteract sarcopenia. If obesity and sarcopenia occur concomitantly, a high-protein diet plus physical exercise (especially resistance training) has been shown as an effective strategy to reduce the loss of musculoskeletal tissue and to improve the quality of life. Although further research is needed, miRNAs can represent not only a useful tool to have a more accurate characterization of the sarcopenic phenotype, but also a potential therapeutic tool to evaluate its status and evolution; more in-depth studies would be needed in particular on man and on an important number of subjects; recently, the work of Kumare Dev et al. [185] showed a positive regulation of a group of miRNAs linked to aging in response to a particular exercise program (sprint interval training). There are other factors that require further research such as (i) the effects of nutritional approaches that maintain an optimal proteostasis (e.g., nutritional/protein timing, ashwagandha), (ii) the anti-inflammatory and antioxidant capacity of some molecules (e.g., *N*-acetylcysteine, omega-3, and vitamin D), (iii) the impact of LCHF (e.g., ketogenic diet), and (iv) the changes in mitochondrial health and function. Therefore, nutritional strategies focusing on the delay of the development of sarcopenia should not only aim to stimulate protein synthesis and inhibit protein breakdown, but also provide molecular tools to enhance mitochondrial function and scavenger inflammation, as well as support healthy aging. Lastly, it is necessary to point out that both primary and secondary sarcopenia probably should not have a defined treatment protocol but should rather be managed through a multidisciplinary team, coordinating the action of nurses, physician, dietitians, kinesiologist, and exercise specialists on a case-by-case basis.

## Figures and Tables

**Figure 1 ijms-22-09724-f001:**
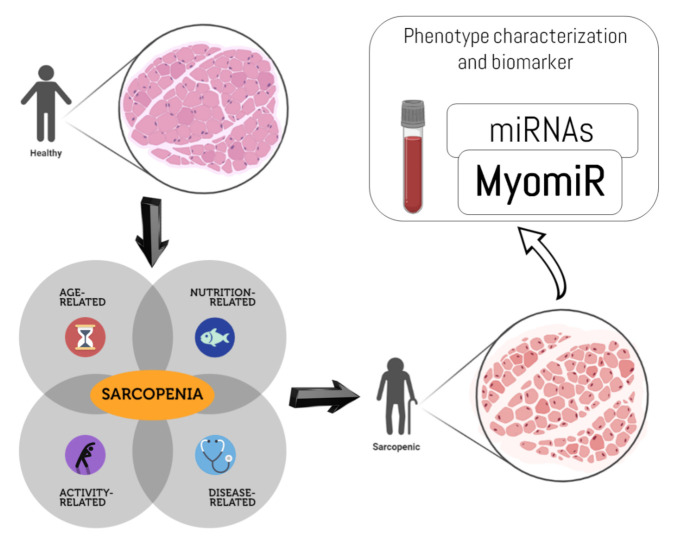
Sarcopenia and the muscle microRNAs (myomiRs). Representation of the main domains of the sarcopenia etiology, the changes in muscle cross-sectional area of lean and intramuscular adipose tissue, and the potential role of the miRNAs secreted by musculoskeletal tissue (myomiRs). Other several neuro-morphophysiological changes (i.e., dysfunction of peripheral nerves, alterations of the neuromuscular junction, modifications of sarcomeric proteins, reduction in the number of the satellite cells, and loss of mitochondrial content and function, among others) are not shown. Source: created by D.A.B. with BioRender—https://biorender.com/ (accessed on 2 September 2021).

**Table 1 ijms-22-09724-t001:** Nutrients and their rationale in sarcopenia treatment.

Nutrient	Rationale and Effects
Amino acids	The branched-chain amino acids (BCAAs), especially leucine, have been shown to play an important role in stimulating muscle protein synthesis through activation of the mTORC1 pathway [114,115]. Nevertheless, it seems that skeletal muscle depends on all essential amino acids rather than BCAAs or leucine alone [115,116,117]. In this sense, a growing body of evidence suggests that essential amino acids might extend healthy life span and prevent pathological conditions associated with an energy deficit (e.g., sarcopenia) [109]. These effects are possibly mediated through mitochondrial biogenesis and the upregulation of antioxidant systems [118].
Ashwagandha	*Withania somnifera* (Ashawagandha), also called “Indian ginseng”, is an herb highly valued and used for centuries by Ayurvedic medicine, mainly for its adaptogenic and antistress activity, which is considered a proven Rasavātam [119] Due to its safety and effectiveness in improving quality of life and physical performance while decreasing fatigability [120], Ashwagandha is one of the most studied herbal products with potential antimicrobial, anti-inflammatory, antitumor, antistress, neuroprotective, cardioprotective, antioxidant, and antidiabetic properties [121,122,123].
HMB	Clinical research has demonstrated the potential muscle protective effects of β-hydroxy-β-methylbutyrate (HMB), a leucine-derived molecule, in conditions that compromise skeletal muscle tissue [124,125,126]. These studies suggest that the p38/MAPK and PI3K/Akt signaling pathways are involved in the anticatabolic effects of HMB in skeletal muscle [123]. In addition to these mechanisms, possibly mechanisms involve inhibition of NF-κB activity, prevention of ROS production, stimulation of muscle cell proliferation and differentiation, inhibition of UPS, inhibition of caspases 3 and 8, and stabilization of the sarcolemma [127,128,129].
Carbohydrates	Although there is no additional benefit of adding carbohydrates to a protein supplement that maximally stimulates muscle protein synthesis [130], muscle glycogen content may affect protein turnover [131]. Insulin is a powerful anabolic hormone that can stimulate not only protein synthesis but also amino-acid transport inside the cell [8]; therefore, sufficient carbohydrate intake may be beneficial for maintaining muscle mass in muscle-wasting conditions such as sarcopenia.
Creatine monohydrate	There are several studies showing that creatine monohydrate supplementation in addition to a strength training protocol can augment muscle mass and function in older adults [132,133,134]. It has been shown that this might be due to energy and mechanical optimization of the cells [135], which results in the prevention of protein degradation [134], an increase in and activation of satellite cells [136,137], and an increase in glycogen synthesis [138]. In addition, potential benefits of creatine outside of musculoskeletal tissue have been demonstrated in the brain, the heart, vascular health, immune system, diabetes, and cancer, among others [138,139,140,141]. Thus, creatine supplementation may have a potential benefit on sarcopenia [142,143,144].
Vitamin D	The elderly population is particularly at risk of vitamin D deficiencies, as their ability to generate precursor molecules in the skin is reduced with the advancement of age [145,146], as well as in obese subjects, probably because it is trapped in the adipose tissue due to low-grade inflammation [147,148,149]. In fact, low levels of vitamin D have been shown to be related to loss of muscle mass, falls, and frailty [150]. Thus, current evidence supports vitamin D supplementation to improve muscle strength [151].

**Table 2 ijms-22-09724-t002:** Protein recommendations based on age and fat-free mass.

	30 to 40 Years Old	40 to 50 Years Old	50 to 60 Years Old	60 to 70 Years Old
No caloric deficit	2.0–2.3 g∙kg^−1^ FFM	2.3–2.6 g∙kg^−1^ FFM	2.6–2.9 g∙kg^−1^ FFM	2.9–3.2 g∙kg^−1^ FFM
Caloric deficit	2.4–2.8 g∙kg^−1^ FFM	2.8–3.1 g∙kg^−1^ FFM	3.1–3.5 g∙kg^−1^ FFM	3.5–3.8 g∙kg^−1^ FFM

## Data Availability

Not applicable.

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
