# Peer review of "Sarcopenia: Etiology, Nutritional Approaches, and miRNAs"

_ijms, 2021, doi:10.3390/ijms22189724_

Round 1

Reviewer 1 Report

In the manuscript titled “Sarcopenia: Etiology, Nutritional Approaches and, miRNAs”, the authors summarized the current knowledge on the metabolic background of sarcopenia and the potential biomarker for assessing sarcopenia progression. They also provide an integral nutritional approach based on the current research. This is an interesting review that would be of interest to the readers. However, I would suggest the authors to provide one or two figures with the potential mechanisms summarized in the main text and a table with the key proteins involved in sarcopenia progression. 

Author Response

Response to Reviewer 1 Comments

In the manuscript titled “Sarcopenia: Etiology, Nutritional Approaches and, miRNAs”, the authors summarized the current knowledge on the metabolic background of sarcopenia and the potential biomarker for assessing sarcopenia progression. They also provide an integral nutritional approach based on the current research. This is an interesting review that would be of interest to the readers. However, I would suggest the authors to provide one or two figures with the potential mechanisms summarized in the main text and a table with the key proteins involved in sarcopenia progression.

Response: We appreciate the reviewer’s comments. According to your suggestion, we have provided a general figure highlighting the potential of myomiR for sarcopenia diagnosis and phenotype characterization. Regarding the table, we have cautiously selected and discussed the key proteins involved in central mechanisms of the sarcopenia progression. In fact, we have organized this information in clearly separated and biologically-specific phenomenons to improve readability; thus, we feel the table would be redundant information.

Reviewer 2 Report

This was a nice review on sarcopenia, which describes also some innovations (as the miRNA, or myomiRNA) on this field. 

In the first section, the Authors discussed definitions of sarcopenia, then pathophysiology and some therapeutic options.

On a general view, the paper is fluent, there are only some typos throughout the manuscript. The number of references is, in my opinion, adequate. 

My questions: 

  • Does the treatment of systemic inflammatory chronic diseases play a role in attenuating sarcopenia? In other words, does remission of inflammatory bowel disease or reumathoid arthritis, as well as the improvement of liver function in cirrhosis, improve sarcopenia? This point, highlighting the underlying chronic disease, should be extensively described.
  • I would add more data about the prognostic role of sarcopenia (e.g., in general surgery, in ICU)
  • What is the role of gender in the interplay between sarcopenia and ageing ? Are women at higher risk of sarcopenia at a younger age than men due to menopause?
  • I agree with the importance of exercise and diet as therapeutic agents for sarcopenia. Nevertheless, exercise and diet cannot be offered in the same way to every patient. For instance, patients with chronic heart failure or with neurological diseases are often frail, bedridden. Similarly, consecutive hospitalizations for acute or chronic conditions may influence the improvement of muscle mass (for instance, patients undergoing hemodialysis, cirrhotic patients requiring several hospitalizations, COPD patients). A comment of this point would be valuable. 
  • In my personal opinion, sarcopenia should be treated through a multidisciplinary team, including for instance nurses, physician, dietitians. Do the Authors agree? A comment would be valuable. 
  • According to my personal experience, the treatment of sarcopenia, especially in patients with underlying chronic diseases, is long and requires multiple settings, both at out- and in-patient settings. 

Author Response

Response to Reviewer 2 Comments

This was a nice review on sarcopenia, which describes also some innovations (as the miRNA, or myomiRNA) on this field.

In the first section, the Authors discussed definitions of sarcopenia, then pathophysiology and some therapeutic options.

On a general view, the paper is fluent, there are only some typos throughout the manuscript. The number of references is, in my opinion, adequate.

Response: We thank the reviewer for positive comment

My questions:

Does the treatment of systemic inflammatory chronic diseases play a role in attenuating sarcopenia? In other words, does remission of inflammatory bowel disease or reumathoid arthritis, as well as the improvement of liver function in cirrhosis, improve sarcopenia? This point, highlighting the underlying chronic disease, should be extensively described.

Response: Thanks for the interesting note, we added a parte on inflammation paragraph

I would add more data about the prognostic role of sarcopenia (e.g., in general surgery, in ICU)

Response: thanks again, it was a part that we didn’t consider but it is important to notice and include in the review

What is the role of gender in the interplay between sarcopenia and ageing ? Are women at higher risk of sarcopenia at a younger age than men due to menopause?

Response: there is no marked difference, as menopause has a greater impact on bone homeostasis but not directly on muscle mass, in our experience more due to the low physical activity that occurs in women especially after 50 years of age

I agree with the importance of exercise and diet as therapeutic agents for sarcopenia. Nevertheless, exercise and diet cannot be offered in the same way to every patient. For instance, patients with chronic heart failure or with neurological diseases are often frail, bedridden. Similarly, consecutive hospitalizations for acute or chronic conditions may influence the improvement of muscle mass (for instance, patients undergoing hemodialysis, cirrhotic patients requiring several hospitalizations, COPD patients). A comment of this point would be valuable.

Response: we agree and we added a comment

In my personal opinion, sarcopenia should be treated through a multidisciplinary team, including for instance nurses, physician, dietitians. Do the Authors agree? A comment would be valuable.

Response: we absolutely agree, in fact probably many pathologies should be treated in a team and sarcopenia should be treated in particular, we added a comment

According to my personal experience, the treatment of sarcopenia, especially in patients with underlying chronic diseases, is long and requires multiple settings, both at out- and in-patient settings.

Response: also in this case, we absolutely agree, our intent is to give support where possible, for example some food supplements such as omega3 can be used for a long time, as specified above instead physical exercise or the ketogenic diet are not for all subjects, we added some comment where needed

we consider these comments particularly useful as it is evident that those who proposed them "experience" sarcopenia in his/her professional life